# The Effect of Rearing and Adult Environment on HPA Axis Responsivity and Plumage Condition in Laying Hens

**DOI:** 10.3390/ani14233422

**Published:** 2024-11-26

**Authors:** Janicke Nordgreen, Lucille Dumontier, Tom V. Smulders, Judit Vas, Rupert Palme, Andrew M. Janczak

**Affiliations:** 1Department of Paraclinical Sciences, Faculty of Veterinary Medicine, Norwegian University of Life Sciences, 1433 Ås, Norway; 2Centre National de la Recherche Scientifique, Institut Français du Cheval et de L’équitation, Institut National de Recherche Pour L’agriculture, L’alimentation et L’environnement, Unite Mixte de Recherche Physiologie de la Reproduction et des Comportements, Universitè de Tours, 37380 Nouzilly, France; lucille.dumontier@inrae.fr; 3Centre for Behaviour & Evolution, Biosciences Institute, Newcastle University, Newcastle Upon Tyne NE1 7RU, UK; tom.smulders@ncl.ac.uk; 4Department of Animal and Aquacultural Sciences, Faculty of Biosciences, Norwegian University of Life Sciences, 1433 Ås, Norway; judit.vas@nmbu.no; 5Experimental Endocrinology, Department of Biological Sciences and Pathophysiology, University of Veterinary Medicine, 1210 Vienna, Austria; rupert.palme@vetmeduni.ac.at; 6Department of Production Animal Clinical Sciences, Faculty of Veterinary Medicine, Norwegian University of Life Sciences, 1433 Ås, Norway; andrew.janczak@nmbu.no

**Keywords:** poultry, rearing, enrichment, stress, corticosterone, feather quality

## Abstract

The relative importance of the early vs. adult environment on the adult phenotype is not fully mapped for all traits. We, therefore, tested the effect of the early (aviary vs. cage) and adult environments (standard vs. additionally enriched furnished cages) on the stress hormone response to restraint in birds that were 35 weeks into lay. Due to the association between stress, fearfulness and feather pecking, we also assessed their plumage condition. No effect of the early environment on stress hormone response to an acute stressor could be detected 35 weeks into lay, but adult enrichment had a favourable effect on overall corticosterone levels. Contrary to prediction, the enriched group had a slightly worse feather score on their bellies, but this did not seem to be associated with feather pecking. These findings add to the body of evidence showing the importance of environmental enrichment and also indicate that while important, early life experience is not always readily detectable in adult responses to challenges.

## 1. Introduction

Early life conditions can have life-long effects on the individual. Laying hens are moved from the rearing to the laying farm at 16 weeks of age. This change of environment makes them well suited for investigating the effects of early life on adult phenotype. Early environmental enrichment can increase robustness [1], while housing in barren conditions may increase fearfulness and decrease cognitive abilities [2,3,4,5]. Furthermore, individuals reared in a complex environment during the early stages of life may be more frustrated by exposure to a barren environment as adults compared to individuals reared in a barren environment [6] 

Globally, the majority of poultry are kept in cages [7], either barren or furnished. Cages give little or no opportunity to move vertically, dustbathe or regulate social encounters. A recent report from the European Food Safety Authority (EFSA) recommends discontinuing the use of cages [8]. Aviary rearing, on the other hand, trains the pullets to navigate vertically, gives more opportunities to perform natural behaviours and avoid aggressive or dominant conspecifics. The variability of the physical and social environment also means birds are intermittently exposed to mild stressors for short durations with the possibility of showing appropriate coping responses. In contrast, life in a barren environment gives little possibility to ‘practice’ responding to mild stressors and novelty, and this may result in long-term increased sensitivity to stress and increased fearfulness. Fearfulness has been associated with feather damage [9,10], and severe feather pecking, when a bird pecks at and pulls out the feathers of a conspecific, is a serious welfare concern for laying hens. Its development may be influenced by the early environment [9,10]. The comparison of birds reared in cages and aviaries indicates that birds reared in cages are more fearful, show less active avoidance behaviour [2,3] and demonstrate poorer cognitive abilities compared to aviary-reared birds [4]. Laying hens that had been subjected to stress early in life showed a stronger corticosterone response to restraint than control birds at 27 weeks of age [11]. However, when birds were tested at 60 weeks of age, the adult, but not the early environment, had a significant effect on the response to a novel object [12].

Being housed in a barren environment may be perceived as stressful, even though it is most likely less intense than the stress induced by traditional chronic stress protocols [1,13,14]. If birds do experience chronic stress, it can have a negative effect both on mental and physical health [15,16]. Physiological effects of chronic stress include negative effects on adult hippocampal neurogenesis (AHN), the immune system, HPA axis functioning and energy metabolism (summarised in [16,17]), which may reduce the ability of the hippocampus to regulate the stress response [18,19]. An increased corticosterone response to acute stress and a slightly elevated baseline would thus be expected, but findings include increased, unchanged and reduced baseline and post-stress hormone levels [13,14,20,21]. Environmental enrichment reduces the consequences of stress and fear [1,22,23,24]. The effect of past and present physical or housing enrichment may thus interact to influence phenotype, and the relative importance of each may change over time. To our knowledge, the long-term effects of the early and adult environment on HPA axis responsivity and plumage condition have not been studied in laying hens. 

We, therefore, tested the effect of different levels of enrichment at rearing and adult housing facilities and their interaction on the physiological stress response and feather cover, a behaviour-related morphological stress indicator. We predicted that birds kept in cages during early life and in adult housing without additional enrichment would have a stronger corticosterone response to restraint and a poorer feather cover than birds kept in an aviary during early life and housed with additional enrichment as adults. We predicted that at 52 weeks of age, the adult environment would have a stronger impact than the rearing (pullet) housing environment. 

## 2. Materials and Methods

### 2.1. Animal Housing and Husbandry

#### 2.1.1. Rearing (Early) Housing Environment

The hens used in this study (N = 256) were part of a larger project for which 384 non-beak-trimmed White Leghorn hens were reared either in a cage (N = 192) or an aviary (N = 192) until they were 18 weeks of age. The animal housing and husbandry have, therefore, also been described previously in [12,25]. The birds were reared at a commercial hatchery and pullet rearing farm (Steinlands & Co., Rogaland, Norway) in one single room measuring 15 m × 72 m (see [12] for a more detailed description). The system consisted of furnished cages measuring 12 m × 0.8 m × 0.6 m (length × height × width) stacked in three tiers. After hatching, chicks were placed on the first and second tiers of the system. The mesh floor of the aviary rows was lined with paper until four weeks of age. From 5 weeks of age, the front of the aviary rows was opened, and the birds could navigate between the different tiers and the floor of the house. The floor of the house was covered with wood shavings. For one of the aviary rows, the front of one tier was kept closed during the whole rearing period. This enclosed space was located in the second tier of the aviary row and contained 250 birds. Thus, they had no access to the floor of the house or the other tiers of the aviary, and this constituted the cage-rearing treatment. From 5 weeks of age, the density was 26 birds/m^2^ for the cage-reared birds and 29 birds/m^2^ for the aviary-reared birds. In the cage and aviary conditions, birds, respectively, had access to 9.6 cm and 3.2 cm of perch space per bird.

All birds were exposed to the same lighting and feeding schedule. Temperature started at 34 °C and was gradually decreased to 19 °C at 16 weeks of age. Birds were exposed to 24 h of light for the first day, followed by a continuous 4:2 light/dark cycle during the first week as recommended by the Lohman LSL management guide. The light schedule was then switched to 16:8 light/dark at two weeks of age and gradually decreased to 9:15 light/dark by 5 weeks of age. Gradual transitions from dark to light and from light to dark were used. Each transition took 20 min.

#### 2.1.2. Laying (Adult) Housing Environment

At 18 weeks of age, the birds were transported 7.5 h by car from the rearing facility to the experimental farm. The henhouse contained 2808 cages organised in 12 rows, with each row containing six tiers. A walkway between the third and fourth tiers formed the second floor of the henhouse. Experimental birds were all housed in the third tier of the second floor, i.e., the top tier. They were housed in social groups of four individuals in two Victorsson T10 furnished cages connected by an opening (15 cm × 18 cm). The opening between the two cages allowed the birds to move freely between the two cages of the cage pair. Each pair of cages containing four birds is hereafter referred to as a cage. Each cage measured 240 cm × 83 cm × 63 cm (width × height × depth), and the four birds sharing a cage came from the same rearing treatment. Each cage was furnished with four perches (75 cm perch space/bird), two nest boxes (1500 cm^2^ each) and a dustbathing platform on the roof of each nest box (750 cm^2^/bird, Figure 1). The treatments were distributed in the henhouse so that cages with birds reared in the aviary were next to cages with birds reared in cages.

All birds were exposed to the same lighting and feeding schedule during their time at the farm. From the age of 18 weeks, they were kept under a 13:11 light/dark cycle and a temperature of 21.1 ± 1.6 °C without exposure to additional daylight from the outside. Gradual transitions from dark to light and from light to dark were used. Each transition took 15 min. Feed and water were provided ad libitum via a food chain running in front of the cages and a water line with nipple drinkers along the back of the cages. For identification purposes, each bird was individually marked by means of a black or white plastic zip-tie around its left or right leg.

The additionally enriched cages that made up the enriched treatment of the adult birds were the same as standard control cages, with the addition of an extra dustbathing tray for stimulating foraging and dustbathing, a hemp pompon to peck at and polyethylene tarp curtains to increase structural complexity. The latter were hung under one of the perches of the cage. In addition, a low-density polyethylene (LDPE) sheet was hung on the upper edge of each opening between the two cage halves. Birds could, therefore, not see past these barriers and either had to move under or around them or push them out of the way to move past them. The extra dustbathing tray (55 cm × 60 cm width × depth with a 2 cm high frame to keep dustbathing material from falling off the tray) was placed on the perches in one half of the cage and refilled weekly with a mixture of feed crumbles and dustbathing pellets made of pelleted wheat husks. The pompon was attached to the cage front above the dustbathing platform so that it hung at the upper half of the cage wall. The enrichment was added to the cages in stages, starting with the platform and polyethylene curtains two weeks after arrival and then the LDPE sheets and pompons at 26 weeks of age (i.e., eight weeks after arrival).

### 2.2. Data Collection

#### 2.2.1. Feather Score

The plumage condition was evaluated as a feather score according to the Welfare Quality^®^ protocol [27] on all birds in each cage (n = 256) at 52 weeks of age. The belly, neck and back were scored for feather cover and given a score of 0, 1 or 2, with 2 being the poorest score.

#### 2.2.2. The Restraint Stress Test

The restraint stress test [28,29] was carried out in September 2020. The birds were then 52 weeks of age and had lived at the laying farm for 35 weeks. The enriched treatment group had thus been subjected to their treatment for approximately eight months. One control and one stress bird were randomly chosen from each cage. For half of the cages, the stress bird was sampled first, and for the other half, the control bird was sampled first, balanced across early life conditions and adult treatment. A baseline blood sample was taken from both birds. The time from opening the cage until securing the blood sample was registered for both birds and used as a covariate in the statistical analysis. The control bird was then released back into the cage, whereas the stressed bird was restrained in a mesh cloth washing bag suspended from a scaffold for ten minutes. After the ten minutes had elapsed, the stressed bird was released from restraint, and a second blood sample was taken from both stress and control. All blood samples were taken by puncturing the brachial vein with a red cannula and collecting the blood with a heparin-coated microvette (Sarstedt, Akershus, Norway). The samples were centrifuged at 2000× *g* for 5 min, and plasma was pipetted off into Eppendorf tubes and immediately frozen on dry ice.

Due to technical difficulties, the final sample size for corticosterone analysis was 232 samples from 118 birds. For 4 of the 118 birds, only one blood sample was taken, whereas the remaining 114 birds had both the before and after samples analysed (details see Table 1). Only birds from which we obtained both samples were included in the statistical analysis.

### 2.3. Corticosterone Analysis

First, 0.5 mL plasma was extracted with 5 mL diethylether, shaken, centrifuged and frozen. Afterwards, the ether phase was transferred into a new glass vial, dried down and re-dissolved in a similar amount of EIA buffer. An aliquot of 50 µL of these extracts was analysed in an in-house corticosterone EIA (for details, see [30]).

### 2.4. Data Handling and Statistical Analysis

All statistical analyses were performed with R, version 4.2.1 [31]. A *p*-value of 0.05 was set as the significance level.

#### 2.4.1. Feather Score

The feather scores for each of the three areas were analysed in separate models. For the feather score on the neck and back, ordinal logistic regression models (clmm from the ordinal package) with cage included as a random effect and early and adult environment and their interaction as fixed effects were applied. *p*-values were calculated by likelihood ratio tests using the ANOVA function from the RVAideMemoire package (2022–2023) [32]. For the belly score, the proportional odds assumption did not hold, and for the feather score for this area, a sum of the individual feather scores for each cage was calculated, and the data were analysed with a linear model with the early and adult environment and their interaction as fixed effects.

#### 2.4.2. Corticosterone Levels

For the corticosterone levels, we used a linear mixed-effects model (LMM) fitted by restricted maximum likelihood estimates using the lmer function from the R package lme4 [33]. The model was checked for assumptions (homogeneity of variances and normal distribution of residuals), and the corticosterone concentration had to be log-transformed to fit the assumptions. The categorical predictors included in the model were hen ID nested within the cage and the plate ID (used for analysis of corticosterone) as random effects, and early environment (cage or aviary), adult environment (enriched or standard adult housing), treatment (stress or control), and time (first and second sample) as fixed effects. All two-way interactions were included in the model. Post-comparisons were analysed using *t*-tests. Because corticosterone levels are known to start rising after 3 min from exposure to handling [34,35], we checked the possible effects of the time to take the first blood sample counted from the moment we opened the cage door to take the first bird out for sampling on corticosterone level. A simple regression model with baseline corticosterone regressed on time to first sample revealed a significant and positive relationship (F_1, 114_ = 5.64; *p* = 0.02); therefore, this covariate was kept in the final model even though it then only tended to be significant.

## 3. Results

### 3.1. Feather Score

Figure 2 gives an overview of the feather score distribution for each area: head, neck and belly.

For the neck, neither the provision of enrichment nor the early environment affected the feather quality (enrichment: *X*^2^ (df = 1, N = 251) = 0.65; *p* = 0.42; early environment: *X*^2^ (df = 1, N = 251) = 1.56; *p* = 0.21). The interaction between them was also not significant (*X*^2^ (df = 1, N = 251) = 0.42; *p* = 0.52). The same was observed for the back (enrichment: *X*^2^ (df = 1, N = 251) = 0.005; *p* = 0.94; early environment: *X*^2^ (df = 1, N = 251) = 0.59; *p* = 0.44; interaction: *X*^2^ (df = 1, N = 251) = 0.04; *p* = 0.83). The feather score for the belly was also not influenced by the interaction of early and adult environment (F_1, 60_ = 0.82; *p* = 0.37) nor by the early environment alone (F_1, 60_ = 2.3; *p* = 0.14). The presence of enrichment in the experimental cages did, however, increase the belly feather score (F_1, 60_ = 6.1; *p* = 0.016), with enriched birds having an average sum of 7.2 (std 2.2), which was higher than the average sum of 5.9 (std 2.0) for birds housed in standard cages.

### 3.2. Corticosterone Levels

Birds housed in additionally enriched cages during the production period had lower corticosterone concentrations than birds housed in standard furnished cages (F_1, 51_ = 4.12; *p* = 0.048) when variance due to the difference between the first and second sample and the stress test had been taken into account. However, the interaction between adult housing and stress treatment was not significant. Stressed birds had significantly higher corticosterone levels compared to control birds in the second plasma sample (post-*t*-test; t_132_ = 2.53; *p* = 0.01), but not in the baseline (*p* > 0.05; F (sample time ∗ treatment group)_1, 111_ = 9.51; *p* = 0.003; Figure 3). Both groups increased significantly from baseline (control group: t_110_ = 6.93; *p* < 0.0001 and stress group: t_111_ = 10.99; *p* < 0.0001 for the paired comparison of pre- and post-test sample within group; F (sample time)_1, 111_ = 162.18; *p* < 0.0001).

Time-to-first sample tended to have an effect on corticosterone concentration (F_1, 71_ = 3.19; *p* = 0.08). The corticosterone concentration (ng/mL) is shown for each timepoint and treatment group in Figure 3.

## 4. Discussion

The current study was designed to increase our understanding of the effects of the early environment on the later stages of life in laying hens. To this end, we tested the effect of cage vs. aviary housing during the rearing phase (early life) and of adult housing with or without additional enrichment on baseline corticosterone, on the corticosterone response to restraint and on feather cover as a phenotypical indicator of stress. We predicted that birds exposed to cage rearing and adult housing without additional enrichment would have a stronger corticosterone response to restraint and a poorer feather cover than birds reared in an aviary with additional enrichment in the home cages as adults. We found no difference in the corticosterone response between the two early life treatments and conclude that so far into lay (52–54 weeks of age, i.e., adult housing environment), the early environment has no detectable influence on baseline HPA axis activity or HPA axis responsivity after an acute stressor. Neither was there any interaction between the early and adult environment, indicating that birds reared in an aviary and kept in standard furnished cages as adults did not have a stronger sensitivity to acute stress than birds reared in an aviary and provided with enrichment as adults. However, when sample time was accounted for, birds housed with enrichment as adults had lower corticosterone levels. We also found that the procedure of blood sampling and handling alone led to a corticosterone increase but that the restraint stress had a significantly stronger effect than handling, reflected in a higher corticosterone increase in the stress group compared to the control group at sample time two. Feather quality was influenced only by the adult environment for one of the body areas scored, namely the belly.

The physical restraint stress test [28,29] can be used to measure both response to acute stress and stress recovery depending on the number and timing of blood samples taken. We sampled at baseline and then after twelve minutes and thus obtained only the approximate peak of the stress response. The corticosterone levels start to increase after about three minutes after the induction of stress [34,35], and ideally, the baseline sample should be taken within that time. We sampled each bird within three minutes of capture, but our results show that the birds’ stress response was initiated by the first opening of the cage and capturing of the first bird to be sampled. As we sampled one stress and one control bird per cage, we could not manage to sample both within three minutes of the cage door opening and controlled for this by including time from the opening of the cage door to the blood sample in our statistical analysis. We also balanced which bird was sampled first across early and adult housing treatments and between the stress and control groups. The stronger response in the stress group compared to the control group confirms that the restraint test was indeed stressful above and beyond being handled and subjected to blood sampling. Thirty-five weeks into lay, when the hens were 52–54 weeks old, no effects of the early environment on corticosterone levels or feather quality could be detected. As enrichment in the laying farm but not the early environment had an effect on overall corticosterone levels, it seems that in adult birds, the current environment has a stronger impact on HPA axis activity than the early environment. This aligns with the results reported in [12], where adult enrichment but not rearing conditions influenced the birds’ response to a novel object at 60 weeks of age. However, the interpretation of the biological importance of the administered enrichment for the stress response is more difficult, as the effect of enrichment was significant only when sample time (before and after stressor) had been accounted for and did not differ between the stress and the control group.

Ericsson and colleagues [11] tested hens in the restraint stress test at 29 weeks of age, following stress administered at 2, 8 and 17 weeks of age, and found a higher corticosterone response to restraint in the birds that had been stressed at eight weeks of age. As in our study, baseline corticosterone was not influenced. However, the administration of stressors over short periods of time is not directly comparable to housing in cages. Studies of housing effects on corticosterone levels show an increase in feather corticosterone in hens housed in conventional cages compared to hens housed in floor pens [36]. Therefore, measures of HPA activity that reflect longer periods could have been a better way of assessing the effects of early environments and adult housing on baseline corticosterone levels.

The enrichment we administered consisted of a dust-bathing platform made out of wood, with elevated edges to keep the dustbathing material on the platform. We administered enough dustbathing material to cover the platform and refilled it once every week. Hens are highly motivated to dustbathe, and they will work to obtain access to it and increase frequency if access has been denied [37]. Interestingly, [38] found no effect of giving access to a dustbathing platform for 6 h per day on plasma corticosterone or on the response to an ACTH injection, even though the platform was frequently used.

The feather quality on different body areas is believed to indicate different welfare challenges: Aggressive pecking centres around the head, feather pecking around the tail and vent pecking occurs close to the vent (see [27] and Figure 1 in [39]). The feather cover on the belly is not that easily interpreted. It is suggested in the Welfare Quality protocol that it is influenced by production and that it can be reduced in highly productive birds. We did not measure egg production, but we have no reason to believe that there were large production differences between hens housed with and without extra enrichment. We did not observe any skin wounds or vent pecking, so the feather loss on the belly did not seem to be caused by injurious pecking. Our impression from observing the hens was that they used the platforms very frequently, often dustbathing all four hens together, being in line with Duncan’s description of dustbathing as a socially facilitated activity [40]. This was, however, not quantified. Possibly, feathers were worn away by frequent use of the wooden shelf for dust bathing, but we cannot say this with absolute certainty. Importantly, the feather cover on the neck and back was not influenced by any of our treatments, indicating no difference in severe feather pecking between early and adult housing treatments.

## 5. Conclusions

In conclusion, when hens were reared in either an aviary or cages and then housed in standard Victorsson T10 furnished cages, or additionally enriched cages as adults, neither the early rearing environment nor the interaction between the rearing and adult environment influenced baseline corticosterone or the acute hormonal stress response. Additional enrichment in adult housing conditions lowered the overall corticosterone levels. The enriched birds had a poorer feather cover on their bellies, but this is probably related to the physical characteristics of the dustbathing shelves and not a consequence of receiving enrichment per se.

## Figures and Tables

**Figure 1 animals-14-03422-f001:**
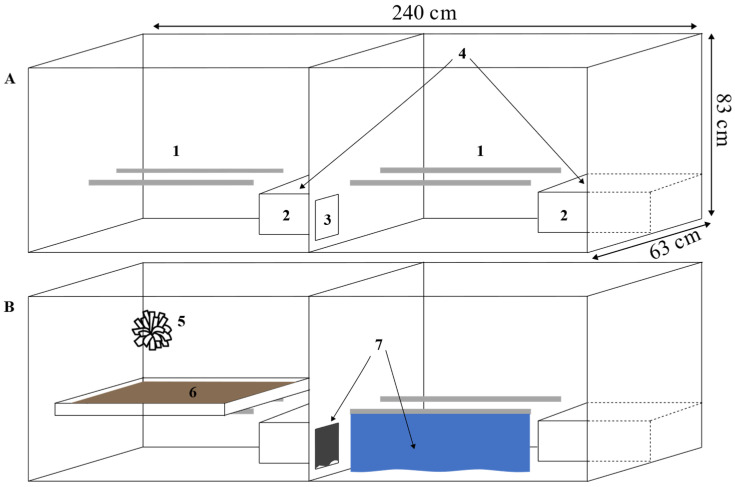
Schemas of the Victorsson T10 furnished cage (**A**) and an additionally enriched cage (**B**), three-quarter front view, showing (1) the perches, (2) the nest boxes, (3) the opening between the two parts of the cage, (4) the dustbathing trays, (5) the hemp pompom, (6) the additional dustbathing tray and (7) the curtains. From Dumontier 2022 [26].

**Figure 2 animals-14-03422-f002:**
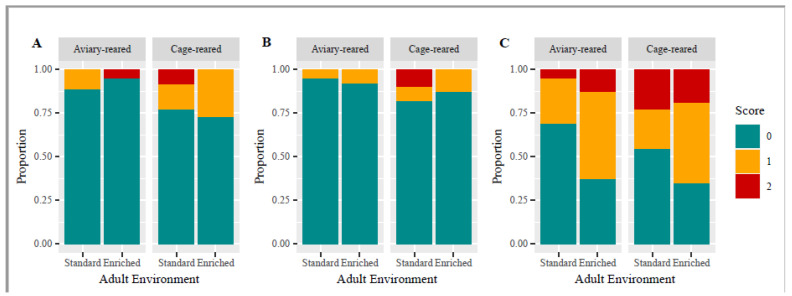
The feather quality score for the neck (**A**), back (**B**) and belly (**C**). Scores range from zero (green bars) to two (red bars), with zero indicating an almost intact plumage and two indicating a very poor condition.

**Figure 3 animals-14-03422-f003:**
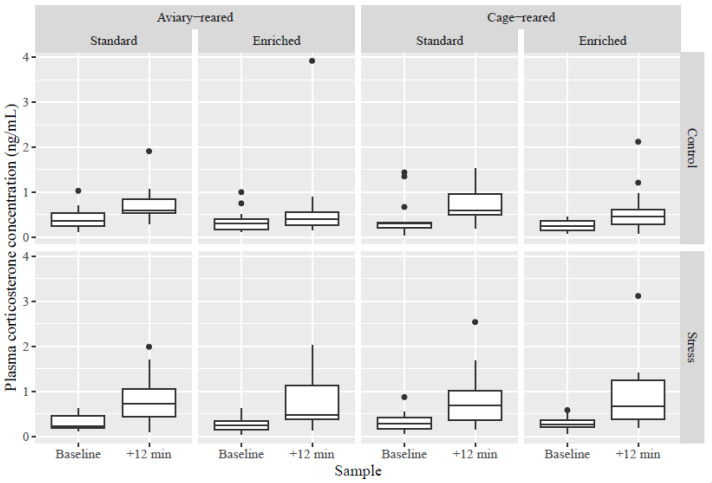
Boxplots of plasma corticosterone concentrations (ng/mL) in control and stressed birds at the first (baseline) and second (+12 min) samples for birds reared in an aviary or in cages and housed in enriched or standard conditions during lay.

**Table 1 animals-14-03422-t001:** The tables give an overview of the number of blood samples, the number of birds with two blood samples (n), the number of birds for which only one sample was secured for the control and the stress group from both adult housing conditions.

**Control Birds**		
**Rearing Housing**	**Adult Housing**	
	Enriched	Standard
Aviary	32 (n = 16)	26 (n = 13)
Cage	31 (n = 15; 1 bird one value, otherwise all birds two samples)	30 (n = 15)
**Stress Birds**		
**Rearing Housing**	**Adult Housing**	
	Enriched	Standard
Aviary	27 (n = 13, 1 bird with one value)	30 (n = 16)
Cage	24 (n = 12)	30 (n = 14; 2 birds with one sample, otherwise all birds two samples)

## Data Availability

The datasets and models will be deposited in an open repository upon publication and can also be made available upon request to the authors.

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
