# Peer review of "The Effect of Rearing and Adult Environment on HPA Axis Responsivity and Plumage Condition in Laying Hens"

_animals, 2024, doi:10.3390/ani14233422_

Round 1

Reviewer 1 Report

Comments and Suggestions for Authors

Dear Authors,

Please see Reviewer response for Animals-3299849 attached.

Regards

Reviewer response for Animals-3299849 – “The effect of rearing and adult environment on HPA-axis responsivity and plumage condition in laying hens”

Dear Authors, very interesting study and intriguing findings. Coping mechanisms are keen for survivability and birds/hens are resilient animals. Please see below some comments and minor details to improve the quality of the manuscript.

Here is one key and important point is that needs do be addressed in the abstract and introduction, likely discussion. Cages can be furnished and non-furnished (conventional). On L 34  - authors mention that cages have little enrichment, but in fact conventional cages have no enrichment at all, so please either name all in this manuscript as furnished cages or clarify this point in line 34 indicating the difference between conventional and  furnished (this with limited EA).

Be consistent with early environment and rearing housing – likewise, be consistent with adult environment and adult housing throughout the manuscript!

There is no information written on statistical analysis in the abstract, a significance value?

L43 – add after lay add (52 weeks of age)

L47 – use stressed group – or even restrained group sounds better?

L53 – please use 52 weeks of age …into lay requires that the person assumes the hens start laying at 18 weeks…not specific…ok in the methods to explain 35 to 37 weeks into lay but not elsewhere.

L59 – read as …at 16 to 18 weeks of age. (add dot) This change of environment ….

L66 – read the majority of egg laying hens are kept in cages (either baren or furnished). Furnished cages …….

L67 - …they give little or no opportunity ….

L84 – how does it influence behaviour?? Be more specific.

L86 – what type of protocol?? Not clear this later portion of the sentence

L95 – read as …past and present physical or housing enrichment….  Add a word along these lines to specify the type of EA you are speaking of.

L99 – read as ….levels of enrichment at rearing and adult housing environment, ….

L105 – replace present by laying or adult environment

L108 – reas as ..Rearing (pullet) housing environment

L110 – WL are mentioned and later, Lohmann guidelines are cited as followed in the laying phase – what is the breed/strain used??? Lohmann is with nn (see L129)

L112 – read as ..hatchery and pullet rearing farm (S…)

L125 – are these measures correct??  Cages had more perch space than in the aviary??

L134 – more information on transport needs to be added! How was it done, how far/long ???conditions of transferring

L133 – read as ...Laying (adult) housing environment

L135 – this Fig 2 is incorrect. Indeed, I strongly recommend that a figure/diagram/infographic or alike is added to the paper to assist readers with understanding the environments and treatments as part of the MM section…this would help and add to the  information stated in Table 1

L151 – replace food by feed, ad libitum italicized; why dimming for 15 min and not as 20 min used during rearing??

L177 – why stating month and year? Not pertinent to the context I believe

L183 – delete brachial vein, keeping it only in L189

L196 – format word ..Figure 2

Table 1 – column A - heading read as Rearing housing as it is stated for adult

L208 – add significance level that was considered as stats diff /p-values??

L233-234 – why is p = 0.02 a tendency??? Please clarify p-value ranges in L208

Figure 1 – if paper printed in grey scale, colors do not stand out very well, suggest using more distinct colors/backgrounds.

Figure 2 – please add on the top of the figure a line that stretches throughout and write “Rearing housing environment” – similarly, on the right side, vertically add a line writing  “Adult housing environment” . The lines help identify these 2 life stages more clearly visually speaking.

L273 – add (adult housing environment) after the word lay

L276 – read as …early environment on stress response into later ….carry over effect on what???

L278 – I suggest using restraint instead of immobilization

L279 – use phenotypic instead of morphological for consistency

L272 – read as …into lay (at 52 to 54 weeks of age)

L302-306 – sentence too long, please add a dot after analysis and state a new sentence.

L338 – write Figure 1 instead of figure one…not clear when reading this part of the sentence.

L339-340 – details on the tray design, material, size etc  should also be stated in the MM, especially because I do agree with the authors, with my long field experience, that the lack of feathers in the belly region is likely due to the dustbathing (as per L344) and not to stress response  - stress related pecking. I could be that the trays were shallow, and hens rubbered the belly against the wood, a more abrasive surface.

Reviewer 2 Report

Comments and Suggestions for Authors

Line 112: “in” is unnecessary.

Lin. 83-84: Does "adult" should mean the age of the bird? The sentence is unclear.

Lin. 144-145:  Under Fig. 2 is something else, not this described in the text, check that.

Given that the experiment is quite complex, a schematic representation of the experiment and/or a photo of the housing would have been useful. In general, the design and methodology of the experiment could be presented more simply.

Line 173: Delete “(2009)”

Line 177: I don't think it is necessary to write when the experiment was carried out.

Line 187: Is it “stressed bird” instead “stress bird” (here and further in the text)?

Lin. 192-194: According to table 1, there are 114 birds from which the samples were taken in enriched housing and 116 in standard housing. Also, there are 112+116=228 blood samples (or 114 complete sets). This is not how is explained in the text. 

Lin. 197-198: The title of a table is usually short. In this case, a description of the table's contents is already given in the paragraph above it. Also, in the text it was said that only those birds from which two samples were taken were taken into account, so it is not necessary to repeat it in the table. However, if table have to remain, the table content would be clearer like this:

Table 1. Number of adult birds for sampling and complete sets of blood samples collected

Control birds

Rearing

Enriched housing

Standard housing

Birds

Complete blood sets

Birds

Complete blood sets

Aviary

32

16

26

13

Cage

31

15*

30

15

Stressed birds

Rearing

Enriched housing

Standard housing

Birds

Complete blood sets

Birds

Complete blood sets

Aviary

27

13*

30

16?

Cage

24

12

30

14*?

* both samples were not taken from all birds

The numbers marked in red do not agree with the number of birds (if there are 30 birds and two samples were taken from all of them, it should be 15 sets instead of 16) and with the text in brackets ("n = 14; 2 birds with one sample, otherwise all birds two samples"; it should be 13 instead 14). Check your results.

Line 214: Delete brackets.

Line 216: “per cage” is unnecessary, delete it.

Line 229: Ad the reference.

Line 238: “Figure 1” should not be bold in description, “three different” is unnecessary, and put colon after “areas”:

Figure 1 gives an overview over the distribution of feather scores for each of the areas: head, neck and belly.

Reviewer 3 Report

Comments and Suggestions for Authors

Thank you for submitting the paper. The effect of rearing and adult environment on HPA-axis responsivity and plumage condition in laying hens. After reading this manuscript, I think this article comprehensively explains and illustrates the effects of early growth environment and current environment on stress response and welfare status of laying hens, which will provide a favorable basis for improving the welfare level of laying hens. However, I personally have some questions and opinions on the article. Please read and refer to:

1. Corticosterone levels can be used as an indicator of HPA (hypothalamic-pituitary-adrenal) axis activation, but measuring corticosterone levels alone is usually not enough to fully prove the activation state of the HPA axis

2. Lines 136-140, the text mentions that two laying hens are raised in each enrichment cage. Why is it necessary to set up a free passage between each cage? If you want the laying hens to move freely in an area the size of two cages, you can completely remove the partition between the two cages. What is the purpose of doing this in the text?

3. Lines 144-145 dustbathing platform on the roof of each nest box (750 cm2/ 144 bird, Fig. 2). fig2 is not found in the text, and fig2 in the text does not match the description in this paragraph.

4. Row 258, figure 2 shows the difference in levels between the stress group and the control group. However, the difference in corticosterone between laying hens in enriched cages and traditional cages is not intuitive, and the results do not indicate whether the comparison is before or after stress.
